# B Cells in Breast Cancer Pathology

**DOI:** 10.3390/cancers15051517

**Published:** 2023-02-28

**Authors:** Mengyuan Li, Angela Quintana, Elena Alberts, Miu Shing Hung, Victoire Boulat, Mercè Martí Ripoll, Anita Grigoriadis

**Affiliations:** 1Cancer Bioinformatics, School of Cancer & Pharmaceutical Sciences, Faculty of Life Sciences and Medicine, King’s College London, London SE1 9RT, UK; 2School of Cancer & Pharmaceutical Sciences, Faculty of Life Sciences and Medicine, King’s College London, London SE1 9RT, UK; 3Vall d’Hebrón Institute of Oncology, 08035 Barcelona, Spain; 4Immunity and Cancer Laboratory, The Francis Crick Institute, London NW1 1AT, UK; 5Immunology Unit, Department of Cell Biology, Physiology and Immunology, Universitat Autònoma de Barcelona, 08193 Barcelona, Spain; 6Biosensing and Bioanalysis Group, Institute of Biotechnology and Biomedicine, Universitat Autònoma de Barcelona, 08193 Barcelona, Spain; 7Breast Cancer Now Unit, School of Cancer & Pharmaceutical Sciences, Faculty of Life Sciences and Medicine, King’s College London, London SE1 9RT, UK

**Keywords:** B cells, tumour-infiltrating lymphocytes, breast cancer, tertiary lymphoid structures, lymph nodes, germinal centres

## Abstract

**Simple Summary:**

B cells in the tumour microenvironment and lymph nodes have affirmed their role in breast cancer pathology. Multiplex imaging, single cell, and spatial transcriptomics of cancer patients’ breast carcinomas and lymph nodes have illustrated the diversity and spatial context of B cells in this disease. Their anti-tumoural and pro-tumoural functions make B cells an attractive research area to improve chemo- and immuno-therapy responses for breast cancer patients.

**Abstract:**

B cells have recently become a focus in breast cancer pathology due to their influence on tumour regression, prognosis, and response to treatment, besides their contribution to antigen presentation, immunoglobulin production, and regulation of adaptive responses. As our understanding of diverse B cell subsets in eliciting both pro- and anti-inflammatory responses in breast cancer patients increases, it has become pertinent to address the molecular and clinical relevance of these immune cell populations within the tumour microenvironment (TME). At the primary tumour site, B cells are either found spatially dispersed or aggregated in so-called tertiary lymphoid structures (TLS). In axillary lymph nodes (LNs), B cell populations, amongst a plethora of activities, undergo germinal centre reactions to ensure humoral immunity. With the recent approval for the addition of immunotherapeutic drugs as a treatment option in the early and metastatic settings for triple-negative breast cancer (TNBC) patients, B cell populations or TLS may resemble valuable biomarkers for immunotherapy responses in certain breast cancer subgroups. New technologies such as spatially defined sequencing techniques, multiplex imaging, and digital technologies have further deciphered the diversity of B cells and the morphological structures in which they appear in the tumour and LNs. Thus, in this review, we comprehensively summarise the current knowledge of B cells in breast cancer. In addition, we provide a user-friendly single-cell RNA-sequencing platform, called “B singLe cEll rna-Seq browSer” (BLESS) platform, with a focus on the B cells in breast cancer patients to interrogate the latest publicly available single-cell RNA-sequencing data collected from diverse breast cancer studies. Finally, we explore their clinical relevance as biomarkers or molecular targets for future interventions.

## 1. Introduction

Breast cancer is the most common cancer in women, with 2.3 million new cases diagnosed per year, attributing to 690,000 deaths annually [1]. Patient outcome is worsened when the tumour disseminates to distant organs, as the five-year overall survival rate in breast cancer patients is reduced from 90% to 29% [1,2]. To date, disease prognosis and treatment stratification are heavily reliant on tumour-, nodal-, and metastasis (TNM) staging and breast cancer subtypes. Based on immunohistochemical (IHC) staining for oestrogen receptor (ER), progesterone receptor (PgR), and human epidermal growth factor receptor 2 (HER2), the majority of the breast cancer cases are classified as ER-positive (~65%), followed by HER2-positive (~20%) and triple-negative breast cancers (TNBC; ~15%) [3].

As a result of their intrinsic molecular heterogeneity, different breast cancer subtypes express varying quantities and assortments of tumour-associated antigens (TAA) and tumour-specific antigens (TSA). These antigens may influence the recruitment and expansion of tumour-infiltrating lymphocytes (TILs) that impact tumour progression. In recent years, the levels of TILs at the primary tumour lesion have been repeatedly shown to correlate positively with better prognosis in TNBCs and HER2-positive cancers [4]. TILs assessment has been demonstrated to be superior to TNM staging when predicting response to chemotherapy, anti-HER2 therapy, and immunotherapy [4,5,6]. Additional clinical observations collectively led to updated recommendations by the St Gallen International Consensus Guidelines 2019 and the upcoming ESMO Early Breast Guidelines 2023. They suggest evaluating stromal TILs in specific carcinomas of TNBC patients, but not to take treatment decisions alone or to escalate or de-escalate treatment. Combining this with stage, age, tumour size, and LN status can help better determine the prognosis [7] thus emphasising the biological relevance of TILs in advancing breast cancer pathology and therapeutic understanding. Due to the current shortage of pathologists and their ever-increasing workload, histological scoring of TILs is currently not performed routinely [8]. To address this discrepancy, several machine-learning algorithms for digital pathology, including the TILs in breast cancer (TIGER) challenge [9], have been developed, in which immune infiltrates at the primary tumour lesion are spatially identified, annotated, and quantified. In the future, these research efforts may streamline the process of TILs scoring and in turn, add valuable information for the treatment regimens for certain breast cancer patients.

Despite these efforts, pathological TILs assessment may, however, not capture the nuances of the immune cell populations present. As such, studies aim to dissect the diverse TILs subsets and their miscellaneous functions in the hope of unveiling additional targetable biomarkers. Tumour-infiltrating T cells (TIL-T) have historically been in the spotlight of tumour immunology networks as they are cardinal in immune-recruitment and eliciting cytotoxic anti-tumour responses. In concordance, the enrichment of intra-tumoral CD4+ helper T cells and CD8+ cytotoxic T cells correlates with a better prognosis and treatment response [10,11,12]. In contrast, many breast tumours exhibit high levels of regulatory T cells (Tregs), which promote the infiltration of immunosuppressive tumour-associated macrophages (TAMs), myeloid-derived suppressor cells (MDSCs), and neutrophils [13]. These cell populations can facilitate TIL-T exhaustion in situ and are associated with an inferior outcome [14]. The diverse functions of immune cell subsets therefore, highlight the necessity to investigate TILs with greater granularity and clarity.

Both TIL-T and tumour-infiltrating B cell (TIL-B) levels demonstrate prognostic value for disease-free and overall survival in cancer, especially for TNBC and HER2-positive breast cancer patients [15]. TIL-B represents around 20% of all immune infiltrates at the primary breast carcinoma, which is elevated compared to normal breast tissue [16,17,18]. High quantities of infiltrating B cells have been reported in around 20% of breast cancers [19], and their levels are highly correlated with the density of TIL-T [15,17]. Within the TME, B cells aid T cell function through the presentation of antigens and secrete antibodies that provide independent anti-tumour cytotoxicity. So far, scoring of immune infiltrates in breast cancer has rarely included B cell populations. Given B cells’ fundamental antigen-presentation capacities, their propensity to secrete anti-tumour antibodies [20], spatial distribution, and their integral role in adaptive responses, B cells are likely to play a crucial role in anti-tumour immunity. As such, the interference of B cells could impact immunotherapeutic approaches [21,22,23], and they must not be overlooked when assessing TILs diagnostically in breast cancer patients. 

## 2. B Cells: Physiological Differentiation, Maturation, and Function

To appreciate the diversity of B cell populations, including their antibody production, maintenance in immunological memory, and their regulation of immune responses, one needs to consider their origin and maturation (Figure 1). The early development of B cells begins in the bone marrow, where haematopoietic stem cells (HSC) (CD34+ CD19−) gradually differentiate to pro-B cells (CD19+ CD10+ CD34+ IgM−), and subsequently pre-B cells (CD19+ CD10+ CD34− IgM−). During this process, immunoglobulin heavy and light chains are synthesised sequentially in the pre-B and pro-B stages, owing to V(D)J recombination. Immunoglobulin chains are later assembled and expressed as the IgM isotype of the B cell receptor (BCR) in immature B cells (IgM+ CD19+ CD20+ CD21+ CD40+). Immature B cells undergo a negative clonal selection for auto-reactivity before exiting the bone marrow and entering the periphery as mature B cells (IgM+ IgD+ CD19+ CD20+ CD21+ CD40+). Chemotactic gradients such as the CCL2/CCR2 and CCR7/CCL19/21 axes recruit circulating B cells into secondary lymphoid organs (SLOs) for further maturation, namely the spleen, tonsils, and LNs. 

After exposure to antigens, naïve B cells within SLOs progressively mature into germinal centre (GC) B cells (CD19+ CD20+ CD27+ CD38+ CD40+ CD83+), memory B cells (CD19+ CD20+ CD21+ CD40+), plasma cells (CD10− CD20− CD38+) or mantle zone (MZ) B cells (IgM+ IgD− CD1+ CD21+) (Figure 1). Antigen-presenting cells (APCs), including dendritic cells and macrophages from the periphery gain access to the SLOs via the subcapsular sinus. Here, they migrate into the T cell zone, where the presentation of cognate antigens drives the differentiation of CD4+ naïve T cells into pre-follicular T helper (pre-Tfh) cells. The cooperation with naïve B cells at the T-B border subsequently induces the final differentiation into T follicular helper cells (Tfh) that will fully activate B cells, directing them towards either an extrafollicular or a GC response, depending on the BCR-antigen affinity. Activated B cells will undergo class switch recombination to express IgA, IgE, and IgG depending on the nature of the antigenic stimuli [24]. B cells that enter GCs will transit between two functionally distinct and polarised areas, the dark zone (DZ) and light zone (LZ), in which somatic hypermutation and affinity maturation take place, respectively [25,26,27] (Figure 1). Somatic hypermutation describes the process by which point mutations accumulate within Ig variable-region encoding genes to generate antibodies. After iterative rounds of proliferation, selection, and tolerance checks, high-affinity B cells will survive and exit the GC as memory B cells and plasma cells, tailored to the nature of the antigenic stimulation. 

Regulatory B cells (Bregs) have been described as an IL-10+ CD1d+ CD5+ CD19+ immunoregulatory B cell population that resemble Tregs in maintaining the balance between self-tolerance and immune activation. The current lack of consensus on Breg-defining markers is also reflected in the ongoing debate with regards to their functions. Bregs primarily secrete cytokines such as IL-10, IL-35, and TGF-β, that restrict the differentiation of pro-inflammatory Th1/Th17 cells [21,22], inhibit the cytotoxic functions of CD8+/NK cells [28], and stimulate the expansion of Tregs [29]. The physiological diversity of B cells, their intertwined activities with other immune and non-immune cells, and their spatially defined development stages, asks for extensive characterisation and quantification of these subsets in the context of breast cancer immune-oncology.

## 3. TIL-B and Adaptive Immunity against Breast Cancer

Distribution of B cell subpopulations within the TIL-B reported in the TME of TNBCs include naïve B cells (~10% of B cells), memory B cells (~80% of B cells), and plasma cells (~20% of B cells) [15,17], suggesting a local presence of adaptive immunity. Over recent years, B cell infiltration has mostly been associated with improved prognosis in breast cancer patients [15,30,31,32,33,34,35]. In particular, memory B cells are significantly enriched in breast cancers compared with healthy tissue [36,37] and are consistently found to be associated with good prognosis in TNBC patients [38]. Upon entering the TME, TIL-B cells encounter both tumour and immune cells, which generates the activation and expansion of specific B cell clones. Cross-linking of the BCR promotes the formation of immune complexes by upregulating BCR signalling pathway molecules *JUN* and *FOS*, lymphocyte activation marker *CD69*, and GC chemokine regulator *RGS1*, which are subsequently associated with a superior outcome in TNBC [34].

In contrast, the presence of intra-tumoral Bregs has been demonstrated as a poor prognostic feature in breast cancer. A study of breast cancer patients in 2019 showed that the abundance of IL10+ Bregs was increased proportionally with Tregs in primary tumours, which coincided with shorter relapse-free disease intervals [39]. This enrichment of Bregs was attributed to the infiltration and regulation by CD33+ MDSCs which contributed to an immunosuppressive TME [40,41]. Similarly, the expansion of IL-10+ CD1d+ CD5+ CD138+ CD19+ Bregs within TNBC may enhance the differentiation of anti-inflammatory M2 macrophages and Tregs to tone down anti-tumour responses [23]. Supporting this, T cells pre-conditioned with Bregs exacerbated lung metastasis in breast cancer xenograft models [42]. The inhibitory activity of Bregs is mediated by CD80-CD86 interactions, the PD-1/PD-L1 axis, in addition to TGF-β and IL-10 production [43]. TGF-β secreted by Bregs has been shown to promote Treg expansion in TNBC mouse models. Upon neoadjuvant chemotherapy treatment, the level of IL-10+ B cells in breast tumours is dramatically reduced and B cells upregulate ICOSL, opting for a more immunostimulatory phenotype. An increase in this ICOSL+ B cell subset is subsequently associated with improved disease-free and overall survival [44]. Although understudied in breast cancer, mouse models and clinical trials in other solid tumours including renal cell carcinoma and melanoma have shown IL-10 inhibition upregulates anti-tumour CD8+ T cell responses and subsequently potentiates immunotherapy treatment [45]. Together, these observations, therefore, present the possibility of targeting Breg functions, thereby preventing immune-cell exhaustion at the primary lesion and distant metastases development.

Studies have further highlighted the importance of CD38+ plasma cell infiltration as an independent factor for progression-free survival and overall survival [46]. Plasma cells in TNBCs exhibit the most significant frequency of IgH somatic hypermutation compared to other TIL-B subsets, with most clones sharing identical IgH sequences [17]. This suggests a high Ig specificity against tumour antigens. However, the rate of plasma cell IgH somatic hypermutation is downregulated in primary TNBC compared to those present in peripheral blood mononuclear cell (PBMC) samples, indicating the potential ability of tumour cells to suppress local humoral response [17]. Given that antibody deficiency in xenograft mouse models accelerates tumour growth [24,47,48,49], it has become apparent that humoral activation by plasma cells is fundamental in halting breast cancer progression.

Immunoglobulin isotype is thought to play an essential role in regulating cancer growth. Profiling immunoglobulins within the TNBC TME indicates that IgG+ TIL-B are more prominent than TIL-B of any other immunoglobulin isotype [17], and IgG isotype switching correlates with a better prognosis [34]. In alignment, the adoptive transfer of tumour-draining LN-derived IgG-secreting plasma cells led to their migration to tumour sites and effectively limited metastatic progression in breast tumour mouse models [24,47,48,49]. An IgG gene expression array has recently been incorporated into the novel HER2DX assay. This signature has been extracted from transcriptomic profiles and patient clinicopathological data of HER2-positive breast carcinoma patients, to estimate the likelihood of recurrence and achieving pathological complete response (pCR) [50,51,52]. Furthermore, over 80% of breast tumour tissues contain either IgG or IgA autoantibodies against a known antigen in a 91-antigen array, namely CTAG1B, MAGEA1, TP53, MUC1, SOX2, BRCA2, TERT [53]. These autoantibodies are rarely detectable in normal adjacent breast tissue and the absence of many of these from patients’ plasma antibody repertoire points to in situ antibody production, presumably by tumour-infiltrating plasma cells. Higher levels of breast-cancer-specific IgG autoantibodies are associated with shorter disease-free survival which is reflected in IgA autoantibodies [53]. Conversely, increased IgG responses are concurrent with lower cytotoxic CD8+ TIL-T and poorer outcomes in some breast cancer patients [16]. IgG antibodies within breast cancer mouse models have been reported to facilitate LN metastasis by targeting the tumour antigen HSPA4. This activates the Src/NF-κB pathway and facilitates metastasis through the CXCR4/SDF-1α axis [11]. More frequently, compelling observations point to tumour-derived IgG having a role in promoting tumour progression [54,55,56], therefore, it remains to be determined whether plasma cells contribute to the secretion of such pathogenic IgG antibodies. A deeper investigation into the contribution of different antibody isotypes on different TIL-B subsets is warranted, especially given the routine use of antibody-based therapies in breast cancer.

High-throughput sequencing technologies have become indispensable to comprehensively capture the molecular features of B cell populations in the TME of breast carcinomas and addressed several aforementioned research gaps. Single-cell (sc)-based signatures can delineate TIL-B subsets more precisely, and have shown superior performance in predicting TNBC patient survival compared to single marker expression e.g., CD20 [17]. Considering the volume of available scRNA-sequencing data and its potential for exponential growth in the B cell field, we have developed a user-friendly “*B singLe cEll rna-Seq browSer*” (BLESS) platform (https://github.com/cancerbioinformatics/BLESS/ (accessed on 19 February 2023)) [36,57,58,59,60,61,62,63,64,65,66,67,68]. BLESS provides the ability to interrogate and visualise human datasets comprised of B cells from primary breast tumours, patient-paired normal breast tissue, PBMCs, and LN samples. By building a comprehensive database (Appendix A), B cell subsets in different immunological sites of breast cancer patients can be investigated pre-, during, or post-treatment. BLESS can delineate their phenotype and their communication networks with adjacent cells, such as endothelial cells, fibroblasts, macrophages, and other TILs in breast tumours. Of note, the source code of BLESS is suitable for sharing and assisting analysis. BLESS’ expandable functionalities provide the opportunity to inspect and compare TIL-B in different clinical settings and will contribute to and elucidate further the multi-functional role of B cells in breast cancer pathology. 

## 4. B Cells in Tertiary Lymphoid Structures

Unique spatial arrangements of B cells in complex cellular contexts have been observed in breast carcinomas by utilising diverse multiplex imaging technologies. B cells in TNBCs are often depleted along the tumour border, and dispersed infiltration of B cells is associated with a lower incidence of recurrence within five years [16,50]. Patients who exhibit heterogeneous clusters containing both B and T cells near cancer cells have superior disease trajectories than those patients who exhibit fewer immune cell aggregates that are larger and further from malignant cell islands [50]. These observations suggest that the distance between immune and tumour cells may potentially influence the capacity of immune-mediated tumour cell killing [69]. 

TLS are immune cell aggregates typically located in the periphery of the tumour and have recently gained attraction in cancer immunology due to their pan-cancer association with a favourable prognosis [70]. In breast cancer patients, the molecular, cellular, and histological presence of TLS has been associated with a better outcome. A 12-chemokine gene signature which can predict the presence of TLS in multiple cancer types was shown to be prognostic for improved survival in breast cancer patients [70]. Moreover, the histological detection of TLS in tumour samples confers improved disease-free survival [71] and overall survival [72] in multiple subtypes of breast cancer. Tumour-infiltrating CXCL13-expressing Tfh cells are closely associated with TLS in breast tumours. CXCL13 is a B cell chemoattractant that selectively binds to CXCR5, and triggers the formation and structural organisation of B cells in TLS. During the maturation of TLS, CXCL13 expression shifts from primarily CD4+ Tfh to CD21+ follicular DC [73]. A gene signature which predicts the presence of these Tfh cells, and CXCL13 expression alone, is prognostic for the survival of untreated breast cancer patients and associated with a higher rate of pCR [74]. TLS are more prevalent in high-grade and early-stage carcinomas, present within as many as 77% of TNBC tumours [11]. TLS are characterised by CD20+ B cells, CD3+ T cells, and mature dendritic cells (DC-LAMP+), however, this can range from disorganised clusters to defined GC-like structures [75]. Additional markers to clarify these distinctions, including Ki67, CD21, CD23, BCL-6, and AID, have identified follicular DCs and Tfh cells within mature TLS, as well as GC-B cell-like centrocyte and centroblast subsets [76,77,78] (Figure 2). These transient structures bear similar morphological characteristics, chemotactic profiles, and B cell maturation features to GCs, with a definitive mantle zone and evidence of polarised chemokine expression [79]. Moreover, the presence of somatically mutated immunoglobulin genes and antibody-producing plasma cells within TLS signifies potentially localised evolution of the B cell response [80].

TLS have also been observed in tumours with an immune excluded or desert phenotype, in which lymphocytes are restricted to the periphery or are completely absent, respectively. This ability to form TLS within immune-cold tumours could indicate some level of immune activation and in turn, a positive influence on the disease trajectory for low TILs breast cancer patients, thus warranting further investigation. Currently, TLS and lymphocytes present within the periphery of the tumour are not included in TIL scoring guidelines recommended by the International TILs Working Group [81]. We hypothesise that for TNBC patients with low TILs, TLS scoring may provide additional information. This is concurrent with recent guidelines from the National Institute for Health and Care Excellence (NICE) recommending immunotherapy as an option in the neoadjuvant and adjuvant setting for locally advanced TNBC patients at high risk of recurrence [82].

## 5. B Cells in Lymph Nodes of Breast Cancer Patients

The initial seeding site for metastatic cells of breast carcinomas is the nearby LN, and as such, the number of involved LNs is a fundamental prognostic factor [83]. However, the role of the LN as an SLO and a site for potential anti-tumour adaptive immune responses has hardly been considered. We have repeatedly shown that the formation of GCs in LNs is associated with a lower risk of developing distant metastasis in TNBC patients with involved LNs, even for TNBC patients with low TILs at their primary lesion [84,85]. Moreover, the frequency of GCs in cancer-free LNs was increased in TNBC compared to non-TNBC patients [85]. As GCs act as the immunological hubs for B cell maturation, this may be reflective of LN-mediated immune responses that are conducive to a superior disease trajectory. Furthermore, patients with more and larger GCs exhibit more frequent TLS formation and higher TILs in their tumours [78], indicating potential crosstalk responsible for the activation of B cells between the primary tumour and the LNs. Thus, the LN as an SLO and its mechanisms to facilitate anti-tumour B cell responses needs to be examined. Morphological features indicative of B cell activation, namely the formation of GC in LNs and TLS in the primary tumour, are ideally suited for robust quantification using deep learning approaches on digitised whole slide images of H&E-stained tumours and LNs. We and others have begun to implement such deep learning frameworks, and demonstrated that an increased frequency of GCs in cancer-free and involved LNs is associated with longer time to distant metastasis in TNBC patients [86]. Tools to automatically annotate TLS in breast carcinomas have not yet been developed. For lung cancer patients, deep learning-based approaches to capture the presence and frequency of TLS have shown their potential as a predictor of tumour recurrence [87,88,89].

Besides the pathological assessments of GCs in LNs, scRNA sequencing studies of LN immune cells have demonstrated both immune-suppressive and immune-promoting roles for B cells [49]. Shariati et al. revealed that an enhanced CD86+CD39+PD-1+PD-L1+ B cell population may hold immunosuppressive properties through association with poor prognostic factors [90], whilst CD40-activated B cells as APCs can activate and expand anti-tumour naïve and memory T cells [91]. Further, interactions between CD45 and the inhibitory receptor CD22 negatively regulated B cell immunity and were reported to be associated with LN metastasis in breast cancer [62]. Tumour-draining LNs may also provide a rich source for tumour-reactive B cells, that give rise to circulating autoantibodies. BCR sequencing in sentinel LNs of breast cancer patients demonstrates hallmarks of affinity maturation against tumour antigen NY-ESO-1 [92]. However, as previously mentioned, tumour-draining LN B cells can also secrete pathogenic IgG that contributes to nodal metastasis in a mouse model of breast cancer [11]. The application of spatial transcriptomics has recently identified ten distinct B cell subtypes spatially distributed across a LN from breast cancer patients, including naïve B cells, activated B cells, GC B cells, and plasma cells [93]. The combination of spatial transcriptomics with scRNA-sequencing data will further deconvolute how these different immune cells may communicate with each other and other cells, and will provide valuable information on B cell subsets and their interaction within GCs, TLS, and the TME of primary breast tumours [94,95]. 

## 6. B Cells Behaviour with Regards to Treatment Responses

By following the same explorative translational studies as TILs, B cell markers and the presence of TLS have been evaluated as potential prognostic and predictive biomarkers in breast cancer patients from clinical trials and translational studies. B cell markers, such as CD20, CXCL13, IgG, and the presence of TLS correlate with a higher pCR rate in neoadjuvant-treated patients (Table 1). This is predominantly associated with improved disease-free survival and overall survival in breast cancer patients receiving a different range of treatment regimens [96,97]. Given the recent approval for the addition of immunotherapeutic drugs as a treatment option in the early and metastatic setting for TNBC, we can explore how different B cell populations are affected by the addition of immunotherapeutic drugs. Pembrolizumab, an antibody targeting PD-1, combined with chemotherapy has been approved to treat early TNBC patients independently of their PD-L1 status [98,99] and as a first-line treatment combined with chemotherapy for advanced TNBC patients expressing PD-L1 (combined positive score [CPS] of at least 10) [100]. Most of the current immunotherapies are developed to target T cells to reverse tumour-mediated exhaustion. However, it remains to be determined whether immune checkpoint inhibitors (ICI) have any direct effect on B cells, and in turn on drug efficacy and patient prognosis. Immunotherapy-related side effects are termed immune-related adverse events (irAEs) and resemble clinical patterns similar to what has been observed in autoimmune diseases whilst being mechanistically distinct [101]. An increase of irAE-related autoantibodies has been observed [102], coupled with a decrease of CD19+ B cells and an increase in CD21^lo^ B cells and plasmablasts [103] in the blood of immunotherapy-treated patients. Conversely, B cell levels are decreased in the organs affected by irAEs, accompanied by high infiltration of CD4+ and CD8+ T cells [102,104,105]. Patients who developed irAEs but did not succumb to these adverse events had an overall better prognosis than those who had not experienced irAEs [106]. B cells are often depleted after chemotherapy, which is potentially a side effect of the treatment and might cause patients to be less responsive to immune targeted therapies [107].

In the BIG 02-98 trial, a higher presence of B cells, particularly memory B cells, was reported in HER2+ and TNBC compared to normal breast tissue, which correlated with higher levels of TILs. They also observed that B cells in the tumour expressed a multitude of cytokines, except for IL-17A, IL-21, and IL-22—which were very low—and higher levels of IFNγ and TNFα in comparison to LN and tonsils [16]. IL-17, IL-21, and IL-22 trigger the creation and assembly of TLS [115,116]. Thus, low levels of those cytokines may hinder or halt their formation. Studies in TNBC mouse models treated with ICI have shown that this will induce the activation of B cells via Tfh to facilitate the anti-tumour response, and subsequently promote the generation of class-switched plasma cells [48]. In breast cancer patients, scRNA-sequencing analyses revealed that responders to combined anti-PD-L1 and chemotherapy treatment exhibited increased baseline levels of intratumoural colocalised B cells and CXCL13+ T cells compared to non-responders. Amongst those patients enriched with B cells, antigen presentation and T cell activation genes are upregulated in responders, in contrast to immunoglobulin production and humoral response genes in non-responders [63]. So far, B cell-associated structures such as GC in LNs, the expression of different immune checkpoints with regards to B cell populations, or other B cell markers in the LNs of patients after treatment have not been elucidated.

PD-1 is distinctively expressed in Tfh cells of GCs in LNs and TLS in primary tumours, which drives the formation of these structures and creates long-lived plasma cells by interacting with PD-L1 and PD-L2 expressed on B cells. Memory B cells have upregulated PD-L2 in comparison to naïve B cells [117]. We have shown a statistically significant higher expression of PD-L2 in the GCs of LNs from TNBC patients with high TIL levels at their primary tumours [78], potentially being associated with an improved long-term prognosis. In the TME, PD-L1 is expressed by tumour cells, macrophages, B cells, and T cells, whilst PD-1 is upregulated on immune cells, mainly T cells. PD-L2 is also present in the malignant epithelial cancer cells of some tumours (but not as widely as PD-L1) and antigen-presenting cells (macrophages and dendritic cells). It is known that anti-PD-1 drugs have more efficacy but also more toxicity than anti-PD-L1 drugs [118]. This distinction may be linked to the difference in expression of these markers not only in the TME but also in the cells of the GC from tumour draining LNs. Antibody-dependent cellular cytotoxicity (ADCC) and antibody-dependent cellular phagocytosis (ADCP) induced by anti-HER2 therapeutics have been well-known for a long time, but recent drugs with the same mechanism have stronger effects on the immune system than the classic anti-HER2 therapies. New anti-HER2 therapies, namely Margetuximab or KN026 enhance ADCC more potently than the combination of trastuzumab and pertuzumab [119,120], whilst BDC-1001, a trastuzumab biosimilar immune-stimulating antibody conjugate (ISAC) chemically conjugated to a toll-like receptor (TLR) 7/8 agonist, has shown promising results [121]. ADCC and ADCP eliminate the tumour mass via cells of the innate immune system, including natural killer cells or macrophages, due to their affinity with the Fc fragment of the antibody on those cells. ADCC and ADCP mechanisms could potentially increase antigen presentation and promote adaptive immunity but not with the same strength of activation as ICIs, which specifically target lymphocytes. This difference in the mechanism of action towards the immune system also translates into differences in the toxicity profile, as anti-HER2 treatments do not cause irAEs. It is probable that the activation of the adaptive immune system follows the natural pathway due to a more indirect effect of these drugs. 

Other promising immunotherapeutic drugs are cancer vaccines, currently under preclinical and clinical development although not approved for breast cancer treatment. The most common modalities for breast cancer are peptide/protein and tumour cell vaccines, and more recently dendritic cell-based and DNA and RNA vaccines. Most of them are directed against the HER2 epitope, and recently developed ones also target HER3, EGFR (epidermal growth factor receptor), VEGF (vascular endothelial growth factor), and IGF-1R (insulin-like growth factor 1). Despite some of these vaccines inducing detectable immune responses with relatively low side effects compared to ICIs and anti-HER treatments, none have demonstrated any significant clinical benefit [122]. 

There are still many pieces missing how B cells can respond to different treatments, interact with other immune cells, and become activated to promote anti-tumour growth. When all these puzzle pieces fit together, we will be able to see the whole picture of the diverse roles of B cells in breast cancer.

## 7. Conclusions

Breast cancer, in particular TNBC, is still an unmet medical need which requires improved treatments with a tolerable safety profile. TILs have consistently been shown to provide prognostic and predictive information, but there are patients with low TILs levels that also show excellent prognoses. Adding B cell markers or the structures in which they take part may add valuable information to estimate responses to different treatments and assess patient outcome. New technologies such as digital pathology, scRNA-sequencing and spatial transcriptomics will undoubtedly play their part in unravelling and exploring known and newly defined B cell populations. These high-throughput methods within a spatial context at the primary tumour site and in the LNs provide exciting entry points as a future research focus in this area.

## Figures and Tables

**Figure 1 cancers-15-01517-f001:**
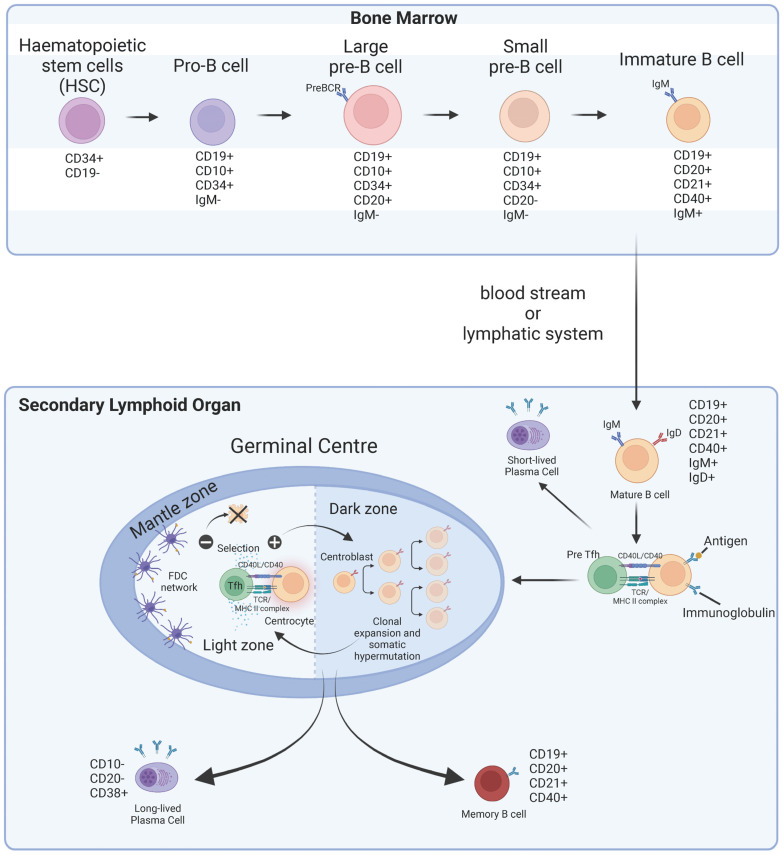
B cell development and differentiation. B cells primarily originate from CD34+CD19− haematopoietic stem cells within the bone marrow, where rearrangement of immunoglobulin heavy and light chains through V(D)J recombination leads to the development of functional IgM+ immature B cells. Those that survive tolerance checks to prevent autoreactivity migrate through the bloodstream or lymphatics to secondary lymphoid organs. Following exposure to cognate antigen, mature B cells can undergo plasmacytic differentiation to generate short-lived plasma cells or establish a germinal centre reaction. Within the germinal centre, B cells undergo iterative rounds of somatic hypermutation and selection through interactions with follicular T helper (Tfh) and follicular dendritic cells (FDC). The resultant B cell pool will exit the germinal centre as high-affinity memory B or plasma cells.

**Figure 2 cancers-15-01517-f002:**
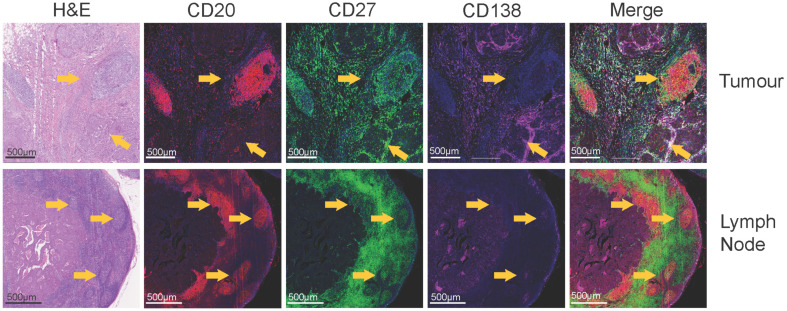
Histological identification of TLS and germinal centres within tumour and LN. The top row displays images of a primary breast tumour with TLS stained with: H&E, CD20, CD27, CD138, and merge of CD20 (red), CD27 (green), and CD138 (magenta). Dense CD20+ B cell areas of the TLS are surrounded by CD27+ cells and dispersed plasma cells. Yellow arrows indicate TLS. The bottom row shows a metastatic LN from the same patient exhibiting germinal centre formation, stained with: H&E, CD20, CD27, CD138, and merge of CD20 (red), CD27 (green) and CD138 (magenta). Yellow arrows indicate germinal centres.

**Table 1 cancers-15-01517-t001:** Summary of B cell markers, related molecules and histological structures, and their prognostic and predictive value observed in breast cancer clinical trials or retrospective studies from 2017 to 2022.

Clinical Trial	# Patients and Cancer Type	Treatment	Prognostic Value	Predictive Value	Reference
BIG 02-98	136 HER2+ and 113 TNBC	Adjuvant CT (not specified) HER2+ patients also received trastuzumab	B cell levels correlated with longer disease-free and overall survival	Not reported	Garaud et al. [16]
CALGB 40601	256 HER2+	Neoadjuvant paclitaxel and trastuzumab +/− lapatinib	Patients with high IGHG counts had a greater 5-year disease-free survival benefit	High IgG-signature (especially the presence of IGHG) showed a significantly higher pCR rate when treated with both anti-HER2 treatments	Fernandez-Martinez et al. [96]
TBCRC006	59 HR+/HR- HER2+	Lapatinib plus trastuzumab HR+ also received endocrine therapy	Not reported	High stromal and intratumoral CD20+ cells were independently associated with a higher pCR rate	De Angelis et al. [97]
Cohort study	30 TNBC	Eribulin	PD-1-neg/PD-1-pos B cell ratio >3 was an independent prognostic factor for worse overall survival	PD-1-neg/PD-1-pos B cell ratio >3 was an independent predictor of shorter response to eribulin (short-term and long-term responders had comparable numbers of B cells levels at their tumours)	Tashireva et al. [108]
Cohort study	368 all breast cancer subtypes (first cohort) 683 all breast cancer subtypes (second cohort) 103 HER2+ cohort	Neoadjuvant CT (doxorubicin + cyclophosphamide followed by paclitaxel, or cyclophosphamide + docetaxel) HER2+ patients also received trastuzumab	B cells induced anti-tumour T cell immunity by upregulating ICOSL and CR2, and downregulating IL-10 after neoadjuvant CT, which is associated with improved prognosis, especially in TNBC	CD55 expression was significantly lower in patients who responded to chemotherapy, which is thought to inhibit the complement and ICOSL	Lu et al. [44]
Cohort study	108 TNBC	Neoadjuvant CT (adriamycin + cyclophosphamide followed by docetaxel)	TLS presence was not associated with overall survival. CXCL13 gene expression correlated with overall survival	HEV density (MECA79+), TLS, B cells, and CXCL13 correlated with each other and with pCR	Song et al. [109]
Cohort study	146 all breast cancer subtypes 71 HR- (validation cohort)	Neoadjuvant CT (cyclophosphamide + epirubicin + fluorouracil (CEF) followed by paclitaxel or docetaxel. Some patients only received CEF, paclitaxel, or docetaxel)	In HR- tumours, high plasma cell infiltration was associated with significantly longer disease-free survival	Tumours with high plasma cells and B cells were associated with pCR There was a higher expression of both PD-1 and PD-L1 on B cells from patients with pCR	Sakaguchi et al. [110]
Cohort study	95 all breast cancer subtypes	Neoadjuvant CT (a combination of different regiments of: adriamycin, (nab)paclitaxel, docetaxel, carboplatin, vinorelbine, and etoposide) +/− bevacizumab (aVEGF) HER2+ patients also received trastuzumab	Not reported	Automate quantitative analysis of CD20 in patient’s tumour independently predicted pCR Patients with tumours displaying high CD20 expression had 5.5 times more pCR	Brown et al. [111]
Cohort study	114 TNBC	Not reported	Stromal CD20 and CD38 correlated with disease-free and overall survival. Intratumoral CD38 and CD138, and stromal CD138 did not correlate with disease-free survival	Not reported	Kuroda et al. [15]
Cohort study	80 DCIS cases (36 pure DCIS and 44 mixed with invasive cancer)	Not reported	Patients with pure DCIS showed a higher number of B cells and lymphoid aggregates than patients with DCIS associated with invasive cancers. Pure DCIS associated with higher numbers of B lymphocytes had shorter recurrence-free interval, but this association was not significant with CD138+ plasma cell count	Not reported	Miligy et al. [112]
Cohort study	102 all breast cancer subtypes	Not reported	Patients with CD20 cells at the primary lesion had better disease-free survival and overall survival	Not reported	Xu et al. [113]
Cohort study	766 TNBC (discovery cohort) 1247 Lung, 1247 colorectal, and 325 melanoma (validation cohort)	Nivolumab (aPD-1) Ipilimumab (aCTLA-4)	Patients with B cells at primary carcinoma had better prognosis in overall survival only when a B cell signature of 9 cytokines (CXCR6, IL18RAP, LCK, IL2RG, CXCL13, PSMB10, TNFRSF14, BATF, TNFRSF4) was low (discovery cohort)	Low levels of the 9 cytokines B cell signature are predictive of response to immunotherapy and have an impact on overall survival (validation cohort)	Lundberg et al. [114]

CT: chemotherapy; DCIS: ductal carcinoma in situ; HER2: human epidermal growth factor receptor 2; HEV: high endothelial venules; HR: hormone receptors; Ig: immunoglobulin; IGHG: Ig heavy chain γ; pCR: pathological complete response; TLS: tertiary lymphoid structure; TNBC: triple-negative breast cancer.

## Data Availability

BLESS (B singLE cell rna-Seq browSer) (https://mengyuanawww.shinyapps.io/shiny_cell_browser/ (accessed on 19 February 2023)) with the source code at GitHub ((https://github.com/cancerbioinformatics/BLESS (accessed on 19 February 2023)) was modified based on https://github.com/yueqiw/shiny_cell_browser (accessed on 19 February 2023).

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
