# Peer review of "B Cells in Breast Cancer Pathology"

_cancers, 2023, doi:10.3390/cancers15051517_

Round 1

Reviewer 1 Report

This review article summarized the fundamental development and roles of B cells, and pathological roles of B cells on breast cancer. 

This article can provide important information on the roles of B cells on breast cancer. 

Author Response

We thank the reviewer for the kind response.

Reviewer 2 Report

Title: B cells in breast cancer pathology.

Authors have written a clear picture of B cell in breast cancer.

1.      Line number 108- 110: The early development of B cells begins in the bone marrow, where haematopoietic stem cells (HSC) (CD34+ CD19-) gradually differentiate to pro-B cells (CD19+ CD10+ CD34+ IgM-), and subsequently pre-B cells (CD19+ CD10+ CD34- IgM-).

Question: Related to the above statement, How do other molecules either upregulate or downregulate while metastasis from benign ?

2.      Authors have written in line number 160-161: The presence of naïve B cells (~10% of B cells), memory B cells (~80% of B cells), and plasma cells (~20% of B cells) have been reported within the TME of HER2-positive breast cancers and TNBCs.

Question: The reference mentioned in this article related to the above statement  have not mentioned the % of B cell types at TME. More over the referenced article had compared with PBMC. Authors suggested to make sure the percentage of B cell types in TNBC.

3.      Line Number 278: Authors are advised to include the reference in the figure description as well.

4.      Line Number 254: Tumour-infiltrating CXCL13-expressing Tfh cells are closely associated with TLS in breast tumours.

Question: Since CXCL13 is a  B lymphocyte chemoattractant, authors suggested to include the formation/ production of CSCL13 and function towards B cell.

5.      How does B cells promote lymph node metastasis through pathogenic IgG?

 6. How does B cell depletion works in breast cancer treatment?

Author Response

We thank the reviewer for the thoughtful comments on our study. We have now revised these sections of our manuscript and have made the following changes.

  1. We are sorry that this B cell section was not clearly explained. Here we discuss the physiological development of B cells and do not refer to metastasis. In the remaining sections of the review, we provide an extensive list of molecules related to B cells, and how they change in breast cancers. To clarify this point, we have changed the title of the section to “B cells physiological differentiation, maturation and function” in line 106

  1. We apologise for the oversight and have now updated the text accordingly. We confirm that figure 3C from reference 17-Garaud et al., eJCI insight 2019, stated the percentage composition of B cells in TNBCs but not HER2+ breast cancers.--This has been corrected in line 162-164 in the following statement: “Distribution of B cell subpopulations within the TIL-B reported in the TME of TNBCs include naïve B cells (~10% of B cells), memory B cells (~80% of B cells), and plasma cells (~20% of B cells)  ”.

  1. This is an original figure created in our lab to depict the three representative markers: CD20, CD27 and CD138 that we agreed on while doing the literature review, as such we cannot add any references. Considering your advice, we have expanded the “Author Contributions” to explain that we created these figures which is addressed in line 460-461 as “staining validation, E.A.”.

  1. We thank the reviewer for this suggestion and have now expanded on CXCL13 and its function towards B cell. We have added the following statements to the text in line 272-274. “Tumour-infiltrating CXCL13-expressing Tfh cells are closely associated with TLS in breast tumours. CXCL13 is a B cell chemoattractant that selectively binds to CXCR5, and their interaction triggers the formation and structural organisation of B cells in TLS. CXCL13 is stimulated by TGFβ and their expression shifts from CD4+ Tfh to CD21+ follicular DC during the maturation of TLS (Ukita et al.,, JCI Insight, 2022).”

  1. We thank the reviewer for this statement and have now added following sentence of IgG’s function in lymph node metastasis promotion in line as “IgG antibodies within breast cancer mouse models have been reported to facilitate LN metastasis by targeting the tumour antigen HSPA4. This activates the Src/NF-kB pathway and facilitates metastasis through the CXCR4/SDF1a axis (Gu et al.,, Nat Med 2019). More frequently, compelling observations point to tumour-derived IgG having a role in promoting tumour progression (Yang et al., PLoS One 2013; Zhang et al.,, Bioengineered 2022, Kdimati et al.,, Int J Mol Sci, 2021), therefore, it remains to be determined whether plasma cells contribute to the secretion of such pathogenic IgG antibodies.”

  1. This is an interesting question. We tried to address this point as we hypothesis that B cell depletion is potentially an adverse side effect of chemotherapy in breast cancer patients. Their depletion may result in an overall depletion of immune cells at the primary tumour site and systemically. We have added following sentence to the text in line 380-382. “B cells are often depleted after chemotherapy, which is potentially a side effect of the treatment and might cause patients to be less responsive to immune targeted therapies (Verma et al.,Breast Cancer Res 2015)”

Reviewer 3 Report

This is an excellent review by Li et al on the role of B cells in breast cancer pathology. The review is well written and organized. The subject matter in the review and its presentation is great.

My only comment is that Table 1 is rather spread out and the spacing could be managed better for a more compact presentation.

Author Response

We thank the reviewer for the positive feedback. We have not updated the format of Table 1. In case there are still formatting issues, we would like to ask the editors on how to best modify the table.

Reviewer 4 Report

Overall Comments: Li and colleagues present a review manuscript that focuses on the role of B cells in breast cancer pathology. Since B cells elicit both a pro- and anti-inflammatory environment in the context of cancer biology, a deep understanding of the role of B cells and their impact on cancer progression is well justified. However, specific examples and references in support throughout the manuscript regarding claims, in particular related to therapeutic impact and prognosis should be added to elevate the impact of an interesting and timely review manuscript. Furthermore, since B cells impact both pro- and anti-growth in cancer, a section highlighting this important aspect and the impact of therapeutics should be added. Once the following comments and questions are addressed, this reviewer believes this review manuscript is suitable for publication in Cancers.

Specific Comments and Questions:

1.)   Abstract: a bit too long and detailed. Please remove the second sentence of Line 25 and the two sentences that follow up to line 31.

2.)   Introduction: lines 100-101: what specific tumor associated antigens do the B cells mount a response against? Please dd specific examples of antigen(s) here.

3.)   Introduction: An intriguing aspect of the review is how B cells elicit a pro- and anti-inflammatory phenotype, this could perhaps be introduced early in the introduction and how a deep understanding of B cell biology in the TME is critical to understand the role(s) of the B cell and how interfering could impact immunotherapeutic approaches.

4.)   In addition to the introduction, a detailed section on how B cells contribute to pro- and anti-tumor growth should be added.

5.)   Additionally, how therapeutics impact cytokine release profiles of B cells in the TME should be described.

6.)   Lines 180-181: How therapeutically could Bregs be impacted in tumor immunology? Are their small molecules that could impact cytokine release profiles? Please add example(s) here.

7.)   Lines 194-195: what isotype(s) indicate a poor prognosis? Please add a description here and relevant reference(s).

8.)   Lines 202-203: what are the known antigen(s) here? Please add, and could also describe the IgM, IgD, and IgE isotypes and impact in the TME if known.

9.)   Related to lines 315-320, does the CD80 co-stimulatory maker expression levels have an impact on immunosuppression?

10.)                  B cells as an anti-cancer vaccine is an active area of immunotherapy. At least a mention of this and an impact on the field will improve the impact.

Author Response

 We thank the reviewer for the thoughtful review and comments on our study. We have fully revised our manuscript and have addressed your comments and made following changes.

  1. The abstract summarizes each section of the review and can give the reader a good sense of what the paper covers. We think it is very important to highlight that we are going to discuss, not only B cells in the TME -including TLS- but also B cells in the LN, which are currently not taken into account into treatment prognosis and response

  1. We thank the reviewer, and found Fridman et al.,, Journal of Experimental Medicine 2020 discussed that “B cells recognize and internalize naïve proteins and glycoproteins via the B cell receptor (BCR)”. Besides from here, we also added several genes as antigens of BCR specific to TIL-B in breast cancer patients later in the main text in “TIL-B and adaptive immunity” section in line 216-219 As “Furthermore, over 80% of breast tumour tissues contain either IgG or IgA autoantibodies against a known antigen in a 91-antigen array, namely CTAG1B, MAGEA1, TP53, MUC1, SOX2, BRCA2, TERT etc (Garaud et al.,, Front Immunol 2018).”

  1. We thank the reviewer, and have added following sentences and references to the review in line 100-105 as “Given B cells’ fundamental antigen-presentation capacities, their propensity to secrete anti-tumour antibodies (Fridman et al., Journal of Experimental Medicine 2020), spatial distribution and their integral role in adaptive responses, B cells are likely to play a crucial role in anti-tumour immunity. As such the interfering of B cells could impact immunotherapeutic approaches (Matsumoto et al.,Immunity 2014; Sun et al., Immunity 2014; Shen et al, Journal of Controlled Release 2015) and they must not be overlooked when assessing TILs diagnostically in breast cancer patients.”

  1. We believe that we do captures these suggestions in several sections and have listed them in detail below:
  • Line 179-181: “Similarly, the expansion of IL-10+ CD1d+ CD5+ CD138+ CD19+ Bregs within TNBC may enhance the differentiation of anti-inflammatory M2 macrophages and Tregs to tune down anti-tumour responses (Shen et al, Journal of Controlled Release 2015)”
  • Line 314-318: “However, the role of the LN as an SLO and a site for potential anti-tumour adaptive immune responses has hardly been considered. We have repeatedly shown that the formation of GCs in LNs is associated with a lower risk of developing distant metastasis in TNBC patients with involved LNs, even for TNBC patients with low-TILs at their primary lesion (Liu et al., NPJ Breast Cancer 2021; Grigoriadis et al., J Pathol Clin Res, 2018).”
  • Line 340-342: “Further, interactions between CD45 and the inhibitory receptor CD22 negatively regulated B cell immunity and were reported to be associated with LN metastasis in breast cancer (Xu et al., Oncogenesis 2021).”

  1. This is an interesting point, and we have added the following statement and reference in line 186-189: “Upon neoadjuvant chemotherapy treatment, the level of IL-10+ B cells in breast tumours is dramatically reduced and B cells upregulate e ICOSL, opting for a more im-munostimulatory phenotype. An increase in this ICOSL+ B cell subset is subsequently associated with improved disease-free and overall survival (Lu et al., Cell 2020)”

  1. We thank the reviewer for this suggestion, and we added following statement in line 186-189 : “Upon neoadjuvant chemotherapy treatment, the level of IL-10+ B cells in breast tumours is dramatically reduced and B cells upregulate e ICOSL, opting for a more im-munostimulatory phenotype. An increase in this ICOSL+ B cell subset is subsequently associated with improved disease-free and overall survival (Lu et al., Cell 2020)”

  1. We thank the reviewer for this suggestion. As we have mentioned, the increase prevalence of IgA might indicate poor prognosis, as well as IgG might promote, under certain conditions, tumour progression. In line 222-226, we mentioned as “Higher levels of breast-cancer specific IgG autoantibodies were associated with shorter disease-free survival which was not the case for IgA autoantibodies (Garaud et al., Front Immunol 2018). Conversely, increased IgG responses are concurrent with lower cytotoxic CD8+ TIL-T and poorer outcomes in some breast cancer patients (Garaud et al., JCI insight 2018)”

  1. We thank the reviewer for this suggestion, and have now added in line 216-219 “Furthermore, over 80% of breast tumour tissues contain either IgG or IgA autoantibodies against a known antigen in a 91-antigen array, namely CTAG1B, MAGEA1, TP53, MUC1, SOX2, BRCA2, TERT etc. (Garaud et al., Front Immunol 2018).”

  1. We thank the reviewer for this suggestion, we found CD80 co-stimulatory marker expression levels have an impact on immunosupression in the TIL-B part from the primary tumour site instead of in lymph nodes. We added the following statements and references in line 183-184: “The inhibitory activity of Bregs is mediated by CD80-CD86 interactions, the PD-1/PD-L1 axis, in addition to TGF-β and IL-10 production (Blair et al., Immunity 2010)”.

  1. We thank the reviewer for this suggestion, we have added the following statement and references to the section “B cells behaviour with regards to treatment responses” in line 436-444:” Other promising immunotherapeutic drugs are cancer vaccines, currently under preclinical and clinical development although not approved for breast cancer treatment. The most common modalities for breast cancer are peptide/protein and tumor cell vaccines, and more recently dendritic cell-based and DNA and RNA vaccines. Most of them are directed against HER2 epitope, and recently developed ones also target HER3, EGFR (epidermal growth factor receptor), VEGF (vascular endothelial growth factor) and IGF-1R (insulin-like growth factor 1). Despite some of these vaccines inducing detectable immune responses with relatively low side effects compared to ICIs and anti-HER treatments, none have demonstrated any significant clinical benefit (Zhu et al., Front Immunol 2022).”

Round 2

Reviewer 4 Report

Li and colleagues present a revised review manuscript that focuses on the role of B cells in breast cancer pathology. Since B cells elicit both a pro- and anti-inflammatory environment in the context of cancer biology, a deep understanding of the role of B cells and their impact on cancer progression is well justified. The authors have significantly improved their manuscript and have satisfactorily addressed this reviewers' concerns. The manuscript is suitable for publication in Cancers.